# Thiosulfinate-Enriched *Allium sativum* Extract Exhibits Differential Effects between Healthy and Sepsis Patients: The Implication of HIF-1α

**DOI:** 10.3390/ijms24076234

**Published:** 2023-03-25

**Authors:** José Avendaño-Ortiz, Francisco Javier Redondo-Calvo, Roberto Lozano-Rodríguez, Verónica Terrón-Arcos, Marta Bergón-Gutiérrez, Concepción Rodríguez-Jiménez, Juan Francisco Rodríguez, Rosa del Campo, Luis Antonio Gómez, Natalia Bejarano-Ramírez, José Manuel Pérez-Ortiz, Eduardo López-Collazo

**Affiliations:** 1Department of Microbiology, University Hospital Ramón y Cajal and Instituto Ramón y Cajal de Investigación Sanitaria (IRYCIS), 28034 Madrid, Spain; jose.avendano@salud.madrid.org (J.A.-O.);; 2Centro de Investigación Biomédica en Red de Enfermedades Infecciosas (CIBERINFEC), Instituto de Salud Carlos III, 28029 Madrid, Spain; 3Department of Anesthesiology and Critical Care Medicine, University General Hospital, 13004 Ciudad Real, Spain; fjredondo@sescam.jccm.es; 4Translational Research Unit, University General Hospital and Research Institute of Castilla-La Mancha (IDISCAM), 13071 Ciudad Real, Spain; 5Faculty of Medicine, Universidad de Castilla-La Mancha, 13071 Ciudad Real, Spain; 6The Innate Immune Response Group, IdiPAZ, La Paz University Hospital, 28046 Madrid, Spain; rob.er_to@hotmail.com (R.L.-R.); vterronarcos@gmail.com (V.T.-A.);; 7Tumor Immunology Laboratory, IdiPAZ, La Paz University Hospital, 28046 Madrid, Spain; 8Department of Chemical Engineering, Institute of Chemical and Environmental Technology, Universidad de Castilla-La Mancha, 13071 Ciudad Real, Spain; juan.rromero@uclm.es (J.F.R.);; 9Department of Pediatrics, University General Hospital, 13004 Ciudad Real, Spain; 10Centro de Investigación Biomédica en Red de Enfermedades Respiratorias (CIBERES), Instituto de Salud Carlos III, 28029 Madrid, Spain

**Keywords:** thiosulfinates, garlic extract, allicin, inflammation, endotoxin tolerance, sepsis, *Allium sativum*, immunomodulation

## Abstract

Garlic (*Allium sativum*) has historically been associated with antioxidant, immunomodulatory, and microbiocidal properties, mainly due to its richness in thiosulfates and sulfur-containing phytoconstituents. Sepsis patients could benefit from these properties because it involves both inflammatory and refractory processes. We evaluated the effects of thiosulfinate-enriched *Allium sativum* extract (TASE) on the immune response to bacterial lipopolysaccharide (LPS) by monocytes from healthy volunteers (HVs) and patients with sepsis. We also explored the TASE effects in HIF-1α, described as the key transcription factor leading to endotoxin tolerance in sepsis monocytes through *IRAK-M* expression. Our results showed TASE reduced the LPS-triggered reactive oxygen species (ROS) production in monocytes from both patients with sepsis and HVs. Moreover, this extract significantly reduced tumor necrosis factor (TNF)-α, interleukin-1β, and interleukin-6 production in LPS-stimulated monocytes from HVs. However, TASE enhanced the inflammatory response in monocytes from patients with sepsis along with increased expression of human leukocyte antigen-DR. Curiously, these dual effects of TASE on immune response were also found when the HV cohort was divided into low- and high-LPS responders. Although TASE enhanced TNFα production in the LPS-low responders, it decreased the inflammatory response in the LPS-high responders. Furthermore, TASE decreased the HIF-1α pathway-associated genes *IRAK-M*, *VEGFA* and *PD-L1* in sepsis cells, suggesting HIF-1α inhibition by TASE leads to higher cytokine production in these cells as a consequence of IRAK-M downregulation. The suppression of this pathway by TASE was confirmed in vitro with the prolyl hydroxylase inhibitor dimethyloxalylglycine. Our data revealed TASE’s dual effect on monocyte response according to status/phenotype and suggested the HIF-1α suppression as the possible underlying mechanism.

## 1. Introduction

For more than 5000 years, garlic (*Allium sativum*) has traditionally been associated with bactericidal, antiviral, anti-inflammatory and antiparasitic properties [1]. Recently, garlic extract and its relative compounds have also shown anticancer effects, such as antiproliferative ability, cell death induction, and reduction of migration and metastasis [2,3,4,5]. One of the most interesting attributes of *Allium sativum* is its antioxidant capacity. This property has been explained by its richness in thiosulfates and sulfur-containing phytoconstituents [1], including allicin, alliin, S-allyl cysteine, vinyldithiins, and flavonoids such as quercetin [1]. Treatment with allicin has been shown to protect against the detrimental effects of nephropathy in a model of diabetes due to its antioxidant capacity and Nrf2 induction [6]. Moreover, this compound reduced reactive oxygen species (ROS) production in murine aortic endothelial cells that had undergone high glucose/hypoxia-induced injury [7].

Regarding the immunomodulatory properties of garlic, most studies have described its anti-inflammatory effects. Garlic has been shown to prevent inflammatory cytokine expression in LPS-activated human whole blood cells, adipocytes, and placental explants [8,9,10]. Some authors have reported that allicin had been shown in vitro to attenuate inflammation by tumor necrosis factor (TNF)-α in epithelial cells and during hypoxia–reoxygenation in cardiomyocytes [11,12]. Moreover, a randomized, double-blind, clinical trial of aged garlic extract supplementation revealed an inflammation reduction in adults with obesity [13]. However, other authors have argued that allicin exhibits an immunomodulatory role rather than overall anti-inflammatory effects, given that this compound had an enhanced proinflammatory response in acute malaria [14], increased TNF-α production in murine macrophages [15], and augmented interferon-γ release in monocyte cultures from patients with vaginitis [16].

In this regard, sepsis patients could benefit from the effects of garlic as this pathology is defined as a dysregulated inflammatory response against an infection. In a recent study, we found that thiosulfinate-enriched *Allium sativum* extract (TASE) used as an adjuvant to ceftriaxone treatment can improve weight and clinical signs (ocular secretions, piloerection) as well as reduce hepatic edema, vacuolization, and inflammation in a rat peritonitis model of sepsis [17]. Nevertheless, it should be noted animal models do not accurately reproduce the complexity of immune response observed in humans [18,19]. According to several authors, sepsis patients exhibit a first phase of systemic inflammation (SIRS) or “cytokine storm” followed by an immunosuppression phase known as compensatory anti-inflammatory response syndrome (CARS). CARS causes a protracted immunosuppression characterized by neutrophilia, increased regulatory T numbers, endotoxin tolerance phenotype in monocytes, high IL-10 and TGF-β levels, and high lymphocyte apoptosis [20,21,22].

Concerning the molecular mechanisms in sepsis, we and others have reported that monocytes from patients with sepsis demonstrated an endotoxin tolerance status in which they are unable to respond to subsequent bacterial stimuli such as lipopolysaccharide (LPS) [23,24,25,26]. This type of cell reprogramming is characterized by low proinflammatory cytokine production, human leukocyte antigen (HLA)-DR expression, increased phagocytosis activity, enhanced tissue remodeling, and impaired antigen presentation [23,24,25,26,27]. According to our previous data, hypoxia inducible factor-1α (HIF-1α) mediated the endotoxin tolerance phenotype by decreasing the cytokine production through the overexpression of IRAK-M pseudokinase and lowering antigen presentation via PD-L1 overexpression [23,24,25,26,27,28,29]. In this line, allicin and other garlic components have been shown to suppress the HIF-1α pathway in human cancer cells [3,4,30]. Nevertheless, the effects of these types of compounds in sepsis monocytes and their phenotype remain unknown.

Herein, we aimed to study the effects of a TASE on monocytes from healthy volunteers (HVs) and patients with sepsis recruited at hospital emergency department admission. In these cells, we analyzed ROS levels, the inflammatory response to the LPS challenge, and the HIF-1α pathway to assess the relevance of the endotoxin tolerance phenotype on the potential immunomodulatory properties of TASE.

## 2. Results

### 2.1. Thiosulfinate-Enriched Allium sativum Extract Reduces Lipopolysaccharide-Triggered Oxidative Stress in Monocytes from Both Patients with Sepsis and Healthy Volunteers

Due to the antioxidant ability of garlic, we tested its potential effects on the oxidative status of monocytes after an inflammatory insult. To do that, we isolated monocytes from both patients with sepsis and HVs (Table 1) and stimulated them with LPS and in combination with TASE. As Figure 1A illustrates, monocytes from patients with sepsis exhibited higher oxidative stress than HV monocytes. Curiously, despite the fact that TASE did not reduce the septic monocytes’ basal oxidative stress levels, it was able to reduce the increase in stress generated by an inflammatory insult in monocytes from both patients with sepsis and HVs (Figure 1B).

### 2.2. Thiosulfinate-Enriched Allium sativum Extract Exhibits Divergent Effects on Monocytes from Patients with Sepsis Compared with Healthy Volunteers

Once we established that TASE prevents an increase in the basal oxidative stress of LPS-challenged monocytes, we studied its effects in the inflammatory response. We analyzed the expression of human leukocyte antigen (HLA)-DR and the production of inflammatory cytokines in response to TASE, LPS, and their combination. Our data indicated that TASE did not exhibit a patent effect on the HLA-DR expression of HV monocytes. In contrast, HLA-DR expression was higher in LPS-challenged septic monocytes in the presence of TASE than in septic monocytes stimulated with LPS alone (Figure 2A). Thus, the effects of TASE on proinflammatory cytokine production after LPS stimulation was revealed to be the opposite in HV monocytes compared with septic monocytes. In addition, whereas TASE induced a decrease in TNF-α, IL-1β, and IL-6 production in LPS-stimulated HV monocytes, it provoked an increment of these cytokines in the septic monocytes (Figure 2B). Other immune markers deregulated in sepsis, such as CD16 or CD206, as well as the anti-inflammatory cytokine IL-10 and the cytokine CXCL-10, did not exhibit a clear pattern (Appendix A).

### 2.3. Thiosulfinate-Enriched Allium sativum Extract Increases the Lipopolysaccharide Response of Monocytes with an Endotoxin-Tolerant Phenotype

Given that TASE exhibited dual immunomodulatory properties in HV and septic monocytes, and sepsis is highly related to an endotoxin-tolerant phenotype in which monocytes poorly respond to LPS [23,24,25,26], we split the HV cohort into two groups: low- and high-LPS responders. The garlic extract induced a significantly high response to LPS in low-responder monocytes, whereas it reduced the response in high-responder monocytes (Figure 3), indicating that TASE improved the response to LPS in endotoxin-tolerant monocytes.

### 2.4. Thiosulfinate-Enriched Allium sativum Extract Reduces HIF-1α Pathway Activation in Sepsis Monocytes

The HIF-1α pathway is one of the key mechanisms involved in the monocyte phenotype during sepsis; thus, we explored the effect of TASE on this pathway. First, HIF-1α activation was analyzed in monocytes from patients with sepsis and HVs. As expected, the septic monocytes exhibited a basal overexpression of the *HIF1A* gene and some of its target genes, including *VEGFα*, *IRAKM*, and *PD-L1* compared with HVs (Figure 4A). It should be noted that no other basal differences between sepsis and healthy monocytes were found in membrane markers or cytokine levels, except for a slight decrease in CD33 in sepsis monocytes (Appendix A). The expression of these HIF-1α-pathway related genes was shown to be reduced in septic monocytes treated with TASE, with a significant reduction of their expression in the LPS-challenged monocytes (Figure 4B–D). In contrast, in monocytes from HVs, TASE did not reveal a clear effect on the expression of these genes, either basally or in combination with LPS (Figure 4B–D). To confirm the involvement of TASE in the modulation of the HIF-1α pathway, we treated HV monocytes with DMOG, a pharmacological HIF-1α inductor. We found that TASE treatment reduced the DMOG-triggered overexpression of the HIF-1α-related genes (Figure 5).

## 3. Discussion

*Allium sativum* has traditionally been associated with beneficial effects in infectious processes due to its antimicrobial, anti-inflammatory, and antioxidant properties [6,17,31,32]. However, it has not yet been introduced in clinical settings. Perhaps its most interesting application could be as an adjuvant to certain treatments [2,17]. In this regard, we and others have evaluated the potential use of garlic and its associated compounds in animal models of endotoxemia or sepsis, thereby revealing its protective effects [17,33,34,35].

Sepsis can lead to an increase in ROS, especially in monocytes [36]. Herein, we found that TASE reduced the increase in oxidative stress caused by the LPS inflammatory insult in both septic and HV monocytes. These data are in accordance with other studies that found the *Allium sativum* derivatives S-allyl cysteine and alliin reduced nitric oxide generation in LPS-treated rats and intracellular ROS in LPS-stimulated THP1-derived macrophages, respectively [33,37].

In a previous study, using the same TASE as a adjuvant, we found TASE reduced inflammation in a rat peritonitis sepsis model [17]. We found different results when we evaluated the effects of TASE on the human monocyte response. Even though the extract decreased the LPS response of HV monocytes, it increased this response in monocytes from patients with sepsis. It should be noted, animal models mainly reproduce the SIRS phase of sepsis and not the complexity of the immune response observed in humans [18,19], which can be also characterized by CARS and immunosuppression [20,21,22]. A reasonable explanation for these divergent effects is that human septic monocytes are locked in an endotoxin-tolerant status, as we observed the same divergent effect of TASE when we split the HV cohort into high (non-tolerant) and low (tolerant) responders. This result suggests that a possible immunomodulatory effect of *Allium sativum* depends on the polarization of the monocytes at that precise moment. This fact should be considered when designing studies to achieve a precision-medicine approach in studies involving garlic or its derivative compounds.

In previous studies, we have described the HIF-1α pathway as the major transcription factor for endotoxin tolerance in monocytes by IRAK-M and PD-L1 upregulation, leading to low cytokine production and decreased antigen presentation, respectively [24,25,26,38]. IRAK-M is a pseudokinase widely known for its role in toll such as receptor signaling inhibition [28,29,38,39,40]. Here, we observed that TASE reduced the expression of both IRAK-M and PD-L1 in LPS-challenged monocytes from patients with sepsis. This result was reproduced in monocytes from HVs when the HIF-1α pathway was induced by the prolyl hydroxylase inhibitor DMOG. Some other studies have proposed regulation of this pathway by garlic-derived compounds and similar antioxidant phytochemicals [41,42]. Along these lines, allicin and diallyl trisulfide suppressed the HIF-1α pathway in cancer and other pathogenic contexts [3,4,7,13,30]. Thus, we propose the suppression of the HIF-1α pathway and the subsequent decrease in IRAK-M as the possible underlying cause of the differential/dual effects of TASE in monocytes.

In our study, septic monocytes were isolated from blood samples taken in the emergency department prior to any treatment. According to Santos et al., this is typically a moment in time with a higher endotoxin-tolerant phenotype and higher ROS levels in septic monocytes [36]. Our data indicate that, at this time, septic patients could benefit from the immunomodulatory properties of *Allium sativum*. Furthermore, a longitudinal study evaluating the antioxidant properties of TASE at various stages of sepsis and in combination with various treatments would be advisable to better define the treatment window and to optimize its use.

In conclusion, we have evaluated the antioxidant and immunomodulatory capacity of a TASE in the context of sepsis and endotoxin tolerance. Our data revealed a differential and dual effect of *Allium sativum* extract in monocytes according to their basal status. TASE had an anti-inflammatory role in non-tolerant monocytes, whereas it had a proinflammatory effect on tolerant ones. Our results indicated that HIF-1α pathway inhibition by TASE in tolerant monocytes appeared to be responsible for this divergent effect. Thus, we have identified a new mechanism to explain the possible immunomodulatory activity of garlic in the context of sepsis. Our data reinforce the idea of a possible therapeutic opportunity for garlic-derived compounds in sepsis, nevertheless further *in vivo* studies are required to demonstrate its possible clinical benefit. 

## 4. Materials and Methods

### 4.1. Study Design and Participants

Patients who met the criteria for sepsis according to the Sepsis-3 definition [43] were recruited from the Emergency Department of La Paz University Hospital; the age- and sex-matched HVs were recruited from the Blood Donor Services of La Paz University Hospital. The clinical characteristics of the patients and HVs are shown in Table 1. Blood samples were collected at admission and prior to any treatment. The exclusion criteria were pregnancy, having received immunosuppressants at admission (i.e., chemotherapy or corticosteroids), and immunodeficiency (primary or acquired). The study was conducted in accordance with the ethical guidelines of the 1975 Declaration of Helsinki and was approved by the La Paz University Hospital Clinical Research Ethics Committee. An informed consent document was obtained from all the included patients and HVs.

### 4.2. Thiosulfinate-Enriched Allium sativum Extract Preparation

The thiosulfinate-enriched *Allium sativum* extract (TASE) was obtained following a patented protocol for the production of lyophilized garlic extract (WO 2008/102036 A1) that is commercially available under the brand name Aliben (Aliben Foods S.L., Ciudad Real, Spain). The thiosulfinate content was determined as described [2], showing 7.03 µg of allicin per mg of lyophilizate. The lyophilizate extract was dissolved in sterile saline (sodium chloride 0.9% from Fresenius Kabi, Barcelona, Spain) to a final stock concentration of 30 mg/mL (≈10 mM allicin). The solution was then filtered with 0.22 µm syringe (JetBiofil, Guangzhou, China). The composition of organic compounds in TASE is shown in Table 2.

### 4.3. Human Monocyte Isolation

Fresh blood from venipuncture was collected in K_2_-EDTA anticoagulant tubes (Vacuette, Greiner BioOne, San Sebastián de los Reyes, Spain) and submitted to Ficoll-Plus (GE Healthcare Bio-Sciences, Piscataway, NJ, USA) gradient according to the manufacturer’s instructions. The layer of peripheral blood mononuclear cells (PBMCs) was washed twice with phosphate-buffered saline (PBS). The monocyte population was enriched by an adherence selectivity protocol [44,45]. Briefly, the percentage of monocytes in total PBMCs was measured by flow cytometry (FACSCalibur, BD Biosciences, Franklin Lakes, NJ, USA), then 500,000 monocytes per well were cultured in Costar 6-well tissue culture-treated plates (Corning, Corning, NY, USA) in Roswell Park Memorial Institute (RPMI) medium (Gibco, Darmstadt, Germany) without serum for 1 h at 37 °C. The non-adherent cells (mainly lymphocytes) were then discarded, and the adhered cells were washed thrice and cultured with RPMI supplemented with 10% fetal bovine serum (Gibco) and 0.01% penicillin and streptomycin mix (Thermofisher, Waltham, MA, USA) at 37 °C in a humidified atmosphere with 5% CO_2_. Monocyte purity was tested by CD14 labeling and flow cytometry analysis (average 71% of CD14^+^ cells).

### 4.4. In Vitro Stimulation

The monocyte cultures from both patients and controls were stimulated with 3 µg/mL of filtered TASE and/or 10 ng/mL LPS from *Escherichia coli* O111:B4 (Merck KGaA, Darmstadt, Germany) for 16 h using the same volume of vehicle (saline) as a negative control. The supernatants were stored at −20 °C until inflammatory cytokine quantification. Cells were divided into 2 tubes for intracellular ROS measurement and flow cytometry characterization.

### 4.5. Cell Reactive Oxygen Species Measurement

To determine intracellular ROS levels, we employed CellROX Green Reagent (Life Technologies, Carlsbad, CA, USA), following the manufacturer’s instructions as previously described [46]. Briefly, 5 µM CellROX reagent was added to the cell cultures 30 min before harvesting, and the cells were co-stained with an anti-human CD14 allophycocyanin antibody (Immunostep, Salamanca, Spain). Samples were acquired with a FACSCalibur (BD Biosciences) and the data were analyzed with FlowJo software (BD Biosciences).

### 4.6. Flow Cytometry Characterization

Monocytes from cell cultures were labeled with a cocktail of Alexa Fluor 488 anti-human CD14 (BioLegend San Diego, CA, USA), BUV496 anti-human HLA-DR (BD Biosciences), BV 510 anti-human CD16 (BioLegend), BV711 anti-human CD206 (BioLegend), BV605 anti-human CD33 (BioLegend), and BUV395 anti-human CD162 (BD Biosciences). True-Stain Monocyte Blocker (BioLegend) reagent was added prior to the label protocol to block the nonspecific binding of some fluorochromes to monocytes. Cells were acquired on a Cytek Aurora Spectral Cytometer (Cytek Biosciences, Fremont, CA, USA) and the data were analyzed employing OMIQ (Dotmatics, Woburn, MA, USA).

### 4.7. Inflammatory Cytokine Quantification

The concentration of interleukin (IL)-1β, TNF-α, CCL2 (MCP-1), IL-6, IL-10, CXCL8 (IL-8), and CXCL10 (IP-10) were quantified in monocyte supernatant by a LEGENDplex HU Essential Immune Response Panel, following the manufacturer’s protocol. Samples were acquired in a FACSCalibur (BD Biosciences), and the data were analyzed with the LEGENDplex Data Analysis Software Suite (Qognit, Inc. San Carlos, CA, USA).

### 4.8. Dimethyloxalylglycine In Vitro Model

To simulate the overactivation of the HIF-1α pathway observed in patients with sepsis, dimethyloxalylglycine (DMOG), a well-known pharmacological inductor of the HIF-1α pathway, was used as previously described [24]. Monocytes were treated with 100 μM DMOG in the presence or not of 3 µg/mL TASE for 16 h. The relative expression of HIF-1α-dependent genes was quantified by real-time quantitative PCR (RT-qPCR).

### 4.9. Relative mRNA Expression Quantification

Total mRNA from monocyte cultures was isolated with the Monarch Total RNA Miniprep Kit (New England Biolabs, Ipswich, MA, USA). The complementary DNA was obtained by reverse transcription of 0.5 μg total RNA using the High-Capacity cDNA Reverse Transcription kit (Applied Biosystems, Foster, CA, USA). The gene expression levels were analyzed by RT-qPCR (Applied Biosystems 7300).

### 4.10. Statistical Analysis

For sepsis vs. control comparisons of quantitative variables, distribution normality was checked by Shapiro–Wilk test in order to define the parametric (*t* test) and non-parametric (Mann–Whitney) analysis. Kruskal–Wallis with Dunn’s post hoc test was used for multiple group comparisons. Paired *t* test was performed to compare the various in vitro stimulation conditions. For qualitative variables, Fisher exact test was used. *p*-values of <0.05 were considered significant at a 95% confidence interval. Statistical analyses were performed with IBM SPSS 23 (Armonk, NY, USA) and GraphPad Prism 8 (San Diego, CA, USA) software. The sample size (*n*) and the statistical test in each analysis are shown in the figure legends. All experiments were performed in at least five biological replicates.

## Figures and Tables

**Figure 1 ijms-24-06234-f001:**
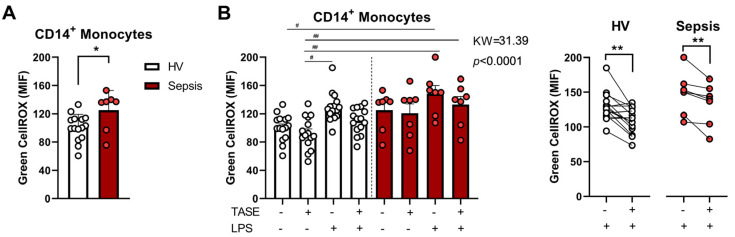
Thiosulfinate-enriched *Allium sativum* extract (TASE) effects on oxidative stress of monocytes from both HV and septic patients. (**A**) Monocytes from HV (*n* = 15) and septic patients (*n* = 7) were labeled with Green CellROX. Mean intensities of fluorescence (MIF) on gated CD14^+^ monocytes determined by flow cytometry are shown. *, *p* < 0.05 in unpaired *t* test. (**B**) Monocytes from HV (*n* = 15) and septic patients (*n* = 7) were stimulated with 10 ng/mL of LPS in the presence or not of 3 µg/mL of TASE for 16 h, then cells were labeled with Green CellROX. MIF on gated CD14^+^ monocytes determined by flow cytometry are shown. KW, Kruskal–Wallis statistic; #, *p* < 0.05; ##, *p* < 0.01 in Dunn’s post hoc test. *, *p* < 0.05; **, *p* < 0.01 in paired *t* test. White dots represent individual HV values and red dots individual sepsis values. Bars express mean ± SD. Left panels show individual paired analyses.

**Figure 2 ijms-24-06234-f002:**
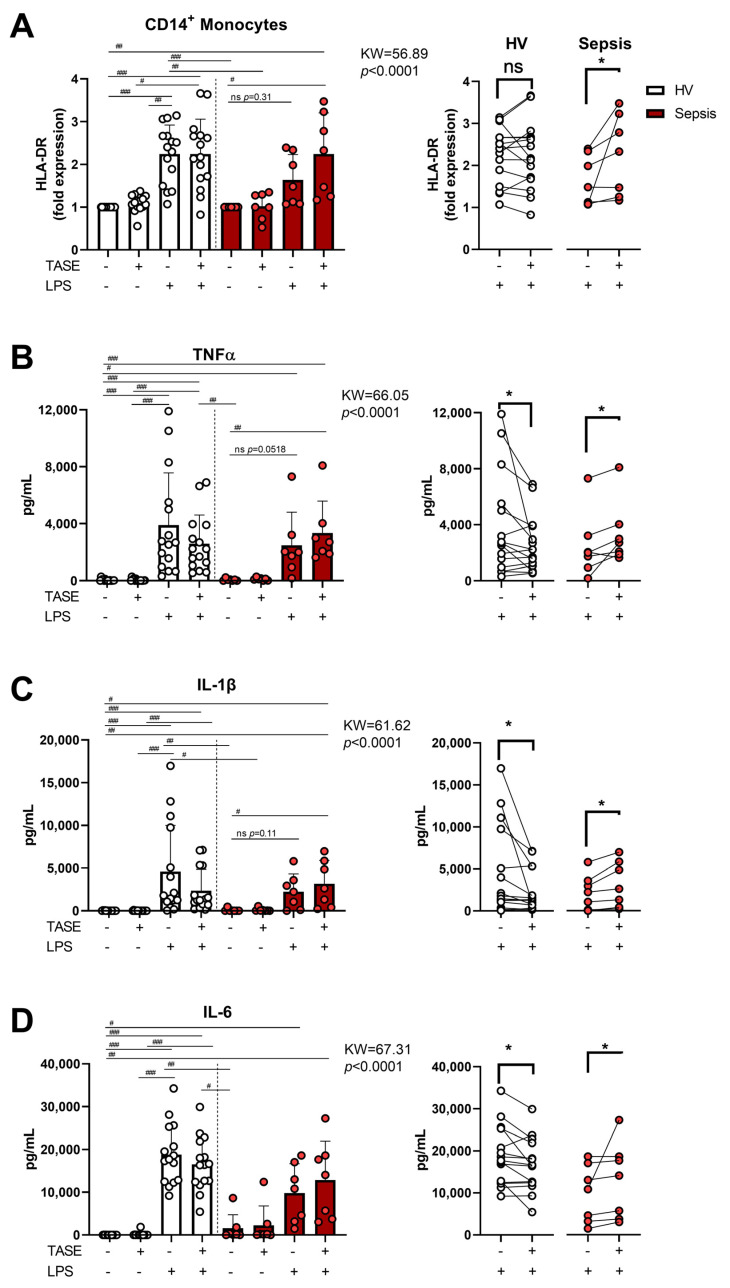
Effects of thiosulfinate-enriched *Allium sativum* extract (TASE) in LPS response of monocytes from both HV and septic patients. (**A**) Monocytes from HV (*n* = 15) and septic patients (*n* = 7) were stimulated with 10 ng/mL for 16 h in the presence or not of 3 µg/mL of TASE and labeled with cytometry antibodies. Fold induction of HLA-DR mean intensities of fluorescence against basal on gated CD14^+^ monocytes determined by flow cytometry are shown. (**B**–**D**) Monocytes from HV (*n* = 15) and septic patients (*n* = 7) were stimulated with 10 ng/mL of LPS for 16 h in combination or not with 3 µg/mL of GE and inflammatory cytokine levels of TNF-α (**B**), IL-1β (**C**) and IL-6 (**D**) were measured. Concentrations are shown. KW, Kruskal–Wallis statistic; #, *p* < 0.05; ##, *p* < 0.01; ###, *p* < 0.001 in Dunn’ post hoc test. *, *p* < 0.05 in paired *t* test. White dots represent individual HV values and red dots individual sepsis values. Left panels show paired analysis of individual values and bars express mean ± SD.

**Figure 3 ijms-24-06234-f003:**
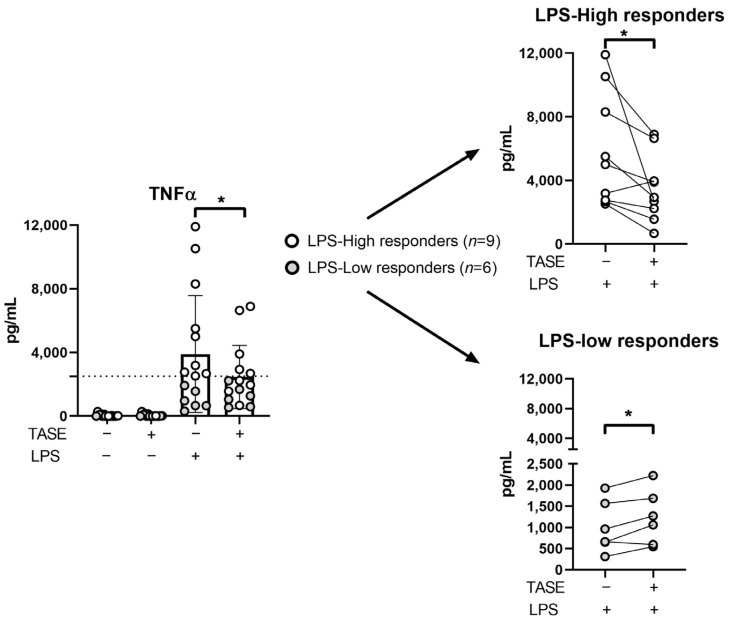
Comparison of thiosulfinate-enriched *Allium sativum* extract (TASE) effects in LPS response of low- and high-responder monocytes. Monocytes from HV (*n* = 15) were stimulated with 10 ng/mL of LPS for 16 h in combination or not with 3 µg/mL of TASE. The HV cohort was split into two groups according to the LPS response of their monocytes (cut-off 2500 pg/mL of TNF-α). TNF-α levels in cell culture supernatant are shown. *, *p* < 0.05 in paired *t* test. White dots represent individual high responders values and grey dots individual low responders values. Left panels show paired analysis of individual values and bars express mean ± SD.

**Figure 4 ijms-24-06234-f004:**
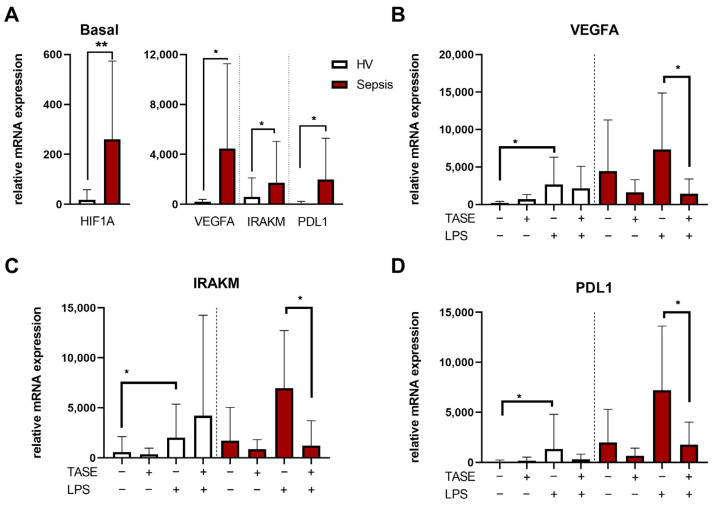
Expression of HIF-1α and related genes in monocytes from HV and septic patients. (**A**) Basal expressions of *HIF1A*, *VEGFA*, *IRAKM,* and *PDL1* in monocytes from HV (*n* = 7) and septic patients (*n* = 7) by RT-qPCR are shown. *, *p* < 0.05; ** *p* < 0.01 in Mann–Whitney *t* test. (**B**–**D**) Monocytes from HV and septic patients were stimulated with 10 ng/mL of LPS for 16 h in combination or not with 3 µg/mL of TASE. Relative expressions of *VEGFA* (**B**), *IRAKM* (**C**), and *PDL1* (**D**) are shown. *, *p* < 0.05 in paired *t* test. Bars represent mean ± SD.

**Figure 5 ijms-24-06234-f005:**
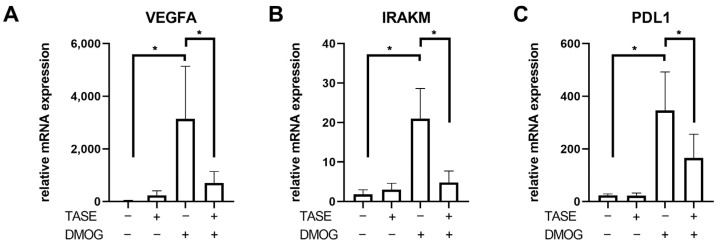
Expression of HIF-1α pathway related genes in monocytes treated with DMOG. Monocytes from HV (*n* = 5) were stimulated with 100 µM of DMOG for 16 h in combination or not with 3 µg/mL of TASE. Relative expressions of *VEGFA* (**A**), *IRAKM* (**B**), and *PDL1* (**C**) are shown. *, *p* < 0.05 in paired *t* test. Bars represent mean ± SD.

**Table 1 ijms-24-06234-t001:** Baseline characteristics of sepsis patients and healthy controls at hospital admission.

	Controls (*n* = 15)	Sepsis (*n* = 7)	*p*-Value
Age—years	51.53 ± 5.21	56 ± 22.8	0.57
Sex, male—*n* (%)	9 (60)	4 (57.1)	0.99
Comorbidities			
Hypertension	1 (6.7)	2 (28.6)	0.23
Diabetes Mellitus	1 (6.7)	1 (14.3)	0.99
COPD	1 (6.7)	2 (28.6)	0.23
Current smoker	2 (15.4)	2 (28.6)	0.56
Temperature—°C	36.5 ± 0.74	37.66 ± 1.6	0.03 *
HR—bpm	84.64 ± 22.7	92.29 ± 18.6	0.44
RR—bpm	15.1 ± 2.5	19.14 ± 4.6	0.03 *
SBP—mm Hg	124.7 ± 33.1	120 ± 43.87	0.78
MBP—mm Hg	86.4 ± 12.3	81.52 ± 24.6	0.54
Leukocytes—10^3^/µL	6.94 ± 1.2	12.2 ± 5.83	0.01 *
Neutrophils—10^3^/µL	4.18 ± 1.33	10.25 ± 5.69	0.01 *
Lymphocytes—10^3^/µL	2.27 ± 0.66	1.18 ± 0.41	<0.01 **
Monocytes—10^3^/µL	0.48 ± 0.13	0.42 ± 0.23	0.25
Platelets—10^3^/µL	236.8 ± 45.1	205.9 ± 88.3	0.72
Creatinine—mg/dL	0.91 ± 0.17	1.54 ± 1.37	0.08
CRP—mg/L	1.35 ± 0.9	136.4 ± 93.5	<0.001 ***
Bilirubin—mg/dL	0.73 ± 0.41	0.42 ± 0.18	0.1
qSOFA	-	0.81 ± 0.73	-

Data are expressed as mean ± SD and percentage (number inside parentheses). COPD, Chronic obstructive pulmonary disease; CRP, C-reactive protein; HR, heart rate; MBP, mean blood pressure; qSOFA, Quick Sequential Organ Failure Assessment score; SBP, systolic blood pressure. *, *p* < 0.05; **, *p* < 0.01; ***, *p* < 0.001 in Chi-square or unpaired *t* tests.

**Table 2 ijms-24-06234-t002:** Composition of organic compounds of lyophilized thiosulfinate-enriched Allium sativum extract (TASE) from purple garlic of Las Pedroñeras (Cuenca, Spain) used in this study.

Compound	TASE (mg/g)
Total Polyphenols	13.91
Total Flavonoids	3.22
Diallyl thiosulfinate (allicin)	7.03
S-allyl-L-cysteine	0.08
Leucine	0.586
Isoleucine	0.500
Valine	0.477
Methionine	0.316
Cysteine	0.811
Phenylalanine	0.556
Tyrosine	4.499
Aspartic Acid	0.901
Glutamic Acid	2.866
Arginine	4.090
Lysine	0.617
Histidine	0.891
Threonine	0.812
Serine	0.385
Glycine	0.215
Alanine	0.897
Thiamine (B1)	0.552
Riboflavin (B2)	0.002
Niacin (B3)	0.026
Pantothenic acid (B5)	1.556
Biotin (B7)	0.251
Cobalamin (B12)	0.898
Ascorbic Acid (C)	3.347
Linoleic Acid (F)	0.276
Tocopherol (E)	0.007
Menadione (K3)	0.007

## Data Availability

Datasheets can be provided by corresponding authors under reasonable request.

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
