# Peer review of "Thiosulfinate-Enriched *Allium sativum* Extract Exhibits Differential Effects between Healthy and Sepsis Patients: The Implication of HIF-1α"

_ijms, 2023, doi:10.3390/ijms24076234_

Round 1

Reviewer 1 Report

March 3, 2023

Manuscript  IJMS-2266052

Thiosulfinate-Enriched Allium sativum extract exhibits differential effects between healthy and sepsis patients: the implication of HIF-1α.

This work is exciting and in the future of being able to be applied in the clinic.

In the abstract, line 31, they say, "Sepsis could benefit" it would not be better to say that it is the patients or people with sepsis who would benefit.

They could justify more why they used the HIF pathway and not another because others could be involved.

On page 2, line 75, again, it is not the pathology that benefits; there are the patients.

On page 2, line 87. What type of blood cells are they referring to?

Can they put more information about sepsis in the introduction? What types of immune cells are activated, and the cytokines are present?

In table 1, there are some comorbidities in the patients they used. Obesity is a low-grade inflammatory state. Hypertensive patients have an activation of innate immunity, including increases in toll-like receptors (TLR) 2 and 4 in peripheral blood monocytes and increased plasma levels of IL-1β and IL-18. Furthermore, inflammation is present during smoking and in respiratory disease.

Doesn't this affect the results?

A graphical abstract gives a final approach to the results.

Author Response

REVIEWER 1

Point 1: This work is exciting and in the future of being able to be applied in the clinic.

Response 1: We want to thank the reviewer for his/her kind comment.

Point 2: In the abstract, line 31, they say, "Sepsis could benefit" it would not be better to say that it is the patients or people with sepsis who would benefit.

Response 2: Following the reviewer recommendation, we have corrected it to “Sepsis patients”.

Point 3: They could justify more why they used the HIF pathway and not another because others could be involved.

Response 3: We agree with the reviewer. In the current version, we have included a more detailed introduction explaining why we have focused on HIF pathway. There are two main reasons why we have studied HIF pathway: 1. We have previously described HIF-1α as a key factor in monocyte reprogramming during sepsis leading to low cytokine production through IRAK-M overexpression.[1,2] 2. Allicin and other garlic components have revealed the ability to suppress the HIF-1α pathway in human cancer cells. [3–5]

Point 4: On page 2, line 75, again, it is not the pathology that benefits; there are the patients.

Response 4: In the current version we have corrected it.

Point 5: On page 2, line 87. What type of blood cells are they referring to?

Response 5: We are referring to monocytes. In current version we have changed.

Point 6: Can they put more information about sepsis in the introduction? What types of immune cells are activated, and the cytokines are present?

Response 6: Following the reviewer’s suggestion, in the current version we have improved our introduction describing the main immune dysregulations in sepsis patients.

Point 7: In table 1, there are some comorbidities in the patients they used. Obesity is a low-grade inflammatory state. Hypertensive patients have an activation of innate immunity, including increases in toll-like receptors (TLR) 2 and 4 in peripheral blood monocytes and increased plasma levels of IL-1β and IL-18. Furthermore, inflammation is present during smoking and in respiratory disease. Doesn't this affect the results?

Response 7: We agree with the reviewer a future deeper study analyzing the thiosulfate-enriched Allium sativa extract (TASE) effects in patients with different comorbidities would be interesting. Nonetheless, as we can see in paired graphs from Figure 1B and Figure 2, sepsis monocytes from all 7 patients included in the study behave practically in the same way (decrease of LPS-induced oxidative stress, increased HLA-DR and cytokine production) indicating TASE effects are similar in all sepsis patients regardless of their comorbidities.

Point 8: A graphical abstract gives a final approach to the results.

Response 8: We thank the reviewer for this comment.

References

  1. Shalova, I.N.; Lim, J.Y.; Chittezhath, M.; Zinkernagel, A.S.; Beasley, F.; Hernández-Jiménez, E.; Toledano, V.; Cubillos-Zapata, C.; Rapisarda, A.; Chen, J.; et al. Human Monocytes Undergo Functional Re-Programming during Sepsis Mediated by Hypoxia-Inducible Factor-1α. Immunity 2015, 42, 484–498, doi:10.1016/j.immuni.2015.02.001.
  2. Avendaño-Ortiz, J.; Maroun-Eid, C.; Martín-Quirós, A.; Toledano, V.; Cubillos-Zapata, C.; Gómez-Campelo, P.; Varela-Serrano, A.; Casas-Martin, J.; Llanos-González, E.; Alvarez, E.; et al. PD-L1 Overexpression During Endotoxin Tolerance Impairs the Adaptive Immune Response in Septic Patients via HIF1α. J. Infect. Dis. 2018, 217, 393–404, doi:10.1093/infdis/jix279.
  3. Wei, Z.; Shan, Y.; Tao, L.; Liu, Y.; Zhu, Z.; Liu, Z.; Wu, Y.; Chen, W.; Wang, A.; Lu, Y. Diallyl Trisulfides, a Natural Histone Deacetylase Inhibitor, Attenuate HIF-1α Synthesis, and Decreases Breast Cancer Metastasis. Molecular Carcinogenesis 2017, 56, 2317–2331, doi:10.1002/mc.22686.
  4. Pandey, N.; Tyagi, G.; Kaur, P.; Pradhan, S.; Rajam, M.V.; Srivastava, T. Allicin Overcomes Hypoxia Mediated Cisplatin Resistance in Lung Cancer Cells through ROS Mediated Cell Death Pathway and by Suppressing Hypoxia Inducible Factors. Cell Physiol Biochem 2020, 54, 748–766, doi:10.33594/000000253.
  5. Song, B.; Shu, Y.; Cui, T.; Fu, P. Allicin Inhibits Human Renal Clear Cell Carcinoma Progression via Suppressing HIF Pathway. Int J Clin Exp Med 2015, 8, 20573–20580.

Reviewer 2 Report

1. There is no clear or justified reasoning in the abstract as to why the inflammatory response is studied alongside targets of the HIF1 pathway. The paper should include studies that investigate pathways that explain how TASE reduces inflammatory response in in LPS-challenged monocytes from healthy people but increases inflammatory response in sepsis patients, particularly if this opposing effect is regulated by HIF1. Similarly, it should include studies of mechanisms that regulate these differential effects of TASE in mouse models of sepsis.

2. Figure 1B, 2A-D, etc, first panel vs second panel: The statistical analysis for each panel should be included.

3. The first reference is missing.

4. Differences between healthy versus sepsis patients (both groups without LPS or TASE) should be tested and significance (or absence of significance) clearly shown throughout the figures.

Author Response

REVIEWER 2

Point 1. There is no clear or justified reasoning in the abstract as to why the inflammatory response is studied alongside targets of the HIF1 pathway. The paper should include studies that investigate pathways that explain how TASE reduces inflammatory response in in LPS-challenged monocytes from healthy people but increases inflammatory response in sepsis patients, particularly if this opposing effect is regulated by HIF1. Similarly, it should include studies of mechanisms that regulate these differential effects of TASE in mouse models of sepsis.

Response 1: We are grateful to the reviewer for the former comments that helped to improve our work. In the current version, we have added a phrase in the abstract and more detailed information in the introduction and discussion explaining why we have focused on HIF-1α pathway. We and others have described IRAK-M pseudokinase inhibit toll like receptor signaling (1–5). In previous reports, we identify HIF-1α causes IRAK-M overexpression by binding to hypoxia response elements in IRAK-M promotor leading to lower cytokine production in sepsis monocytes (6,7). Our data here, reveals HIF-1α related genes, including IRAK-M, are overexpressed in sepsis monocytes and TASE decreased their expression to near healthy volunteer levels. In addition, we have modeled the HIF-1α overactivation in healthy monocytes by DMOG treatment finding TASE also decreases the expression of IRAK-M.

Regarding animal models, we performed a previous study using the same TASE in a rat model of sepsis (8). In this model we found TASE decreased inflammatory cytokines. The explanation of these divergent results could be animal models mainly reproduce the SIRS acute phase of sepsis and not the complex immune response observed in humans (9,10), also characterized by CARS, endotoxin tolerance and immunosuppression (11–13). We prefer to use in vitro/ex vivo stimulation in patient cells to study innate immune response. In the current version, we have added a brief discussion about this.

Point 2. Figure 1B, 2A-D, etc, first panel vs second panel: The statistical analysis for each panel should be included.

Response 2: We want to clarify right panels with paired values are only shown for better visualization of individual TASE effects. The analysis in 1B, 2A-D was paired t-test in both left and right panels. We used paired analysis since we aimed to compare conditions. In this manner, we have focused on comparing the individual response of monocytes from the same healthy subject when stimulated with LPS vs when stimulated with LPS+TASE. Monocytes may respond high or low depending on the donor and increasing data dispersion, so paired analysis minimizes the batch effect when dealing with monocytes from different healthy donors.

Point 3. The first reference is missing.

Response 3: We thank the reviewer. In the current version, we have added this reference.

Point 4. Differences between healthy versus sepsis patients (both groups without LPS or TASE) should be tested and significance (or absence of significance) clearly shown throughout the figures.

Response 4: We thank the reviewer for this insightful recommendation. In current version, we have added a basal (untreated) comparison between sepsis and healthy of all analyzed markers and cytokines in Supplementary Figure 3. Other healthy vs sepsis comparisons were already in the manuscript including comorbidities (Table 1), oxidative stress (Figure 1A), and HIF1A, VEGFA, IRAKM and PDL1 gene expression (Figure 4A).

References

  1. Maldifassi MC, Atienza G, Arnalich F, López-Collazo E, Cedillo JL, Martín-Sánchez C, et al. A New IRAK-M-Mediated Mechanism Implicated in the Anti-Inflammatory Effect of Nicotine via α7 Nicotinic Receptors in Human Macrophages. PLoS One. 2014;9(9):e108397.
  2. del Fresno C, Soler-Rangel L, Soares-Schanoski A, Gómez-Piña V, González-León MC, Gómez-García L, et al. Inflammatory responses associated with acute coronary syndrome up-regulate IRAK-M and induce endotoxin tolerance in circulating monocytes. J Endotoxin Res. 2007;13(1):39-52.
  3. Escoll P, del Fresno C, García L, Vallés G, Lendínez MJ, Arnalich F, et al. Rapid up-regulation of IRAK-M expression following a second endotoxin challenge in human monocytes and in monocytes isolated from septic patients. Biochem Biophys Res Commun. 2003;311(2):465-72.
  4. Kobayashi K, Hernandez LD, Galán JE, Janeway CA, Medzhitov R, Flavell RA. IRAK-M is a negative regulator of Toll-like receptor signaling. Cell. 2002;110(2):191-202.
  5. van ’t Veer C, van den Pangaart PS, van Zoelen MAD, de Kruif M, Birjmohun RS, Stroes ES, et al. Induction of IRAK-M is associated with lipopolysaccharide tolerance in a human endotoxemia model. J Immunol. 2007;179(10):7110-20.
  6. Shalova IN, Lim JY, Chittezhath M, Zinkernagel AS, Beasley F, Hernández-Jiménez E, et al. Human monocytes undergo functional re-programming during sepsis mediated by hypoxia-inducible factor-1α. Immunity. 2015;42(3):484-98.
  7. Avendaño-Ortiz J, Maroun-Eid C, Martín-Quirós A, Toledano V, Cubillos-Zapata C, Gómez-Campelo P, et al. PD-L1 Overexpression During Endotoxin Tolerance Impairs the Adaptive Immune Response in Septic Patients via HIF1α. J Infect Dis. 2018;217(3):393-404.
  8. Redondo-Calvo FJ, Bejarano-Ramírez N, Baladrón V, Montenegro O, Gómez LA, Velasco R, et al. Black Garlic and Thiosulfinate-Enriched Extracts as Adjuvants to Ceftriaxone Treatment in a Rat Peritonitis Model of Sepsis. Biomedicines. 2022;10(12):3095.
  9. Schroder K, Irvine KM, Taylor MS, Bokil NJ, Le Cao KA, Masterman KA, et al. Conservation and divergence in Toll-like receptor 4-regulated gene expression in primary human versus mouse macrophages. Proc Natl Acad Sci USA. 2012;109(16):E944-953.
  10. Seok J, Warren HS, Cuenca AG, Mindrinos MN, Baker HV, Xu W, et al. Genomic responses in mouse models poorly mimic human inflammatory diseases. Proc Natl Acad Sci USA. 2013;110(9):3507-12.
  11. Hotchkiss RS, Monneret G, Payen D. Sepsis-induced immunosuppression: from cellular dysfunctions to immunotherapy. Nat Rev Immunol. 2013;13(12):862-74.
  12. Hotchkiss RS, Coopersmith CM, McDunn JE, Ferguson TA. The sepsis seesaw: tilting toward immunosuppression. Nat Med. 2009;15(5):496-7.
  13. Cao C, Yu M, Chai Y. Pathological alteration and therapeutic implications of sepsis-induced immune cell apoptosis. Cell Death Dis. 2019;10(10):1-14

Round 2

Reviewer 2 Report

Paired or unpaired t-tests or student's t-tests are innappropriate statistical tests for the left panels in 1B and 2A-D.

Author Response

Point 1: Paired or unpaired t-tests or student's t-tests are innappropriate statistical tests for the left panels in 1B and 2A-D.

We thank the reviewer for this comment. Following her/his suggestion, in current version we have done a Kruskal-Wallis with Dunn's post-hoc test for multiple group comparison in the left panels of 1B and 2A-D. In adition, minor english errors have been corrected.